# The importance of individual beliefs in assessing treatment efficacy

Luisa Fassi[1,2,3]*[†], Shachar Hochman[4†], Zafiris J Daskalakis[5], Daniel M Blumberger[6], Roi Cohen Kadosh[3,4]*

[1]MRC Cognition and Brain Sciences Unit, University of Cambridge, Cambridge, United Kingdom; [2]Department of Psychiatry, University of Cambridge, Cambridge, United Kingdom; [3]Department of Experimental Psychology, University of Oxford, Oxford, United Kingdom; [4]School of Psychology, University of Surrey, Surrey, United Kingdom; [5]Department of Psychiatry, University of California, San Diego, San Diego, United States; [6]Temerty Centre for Therapeutic Brain Intervention at the Centre for Addiction and Mental Health and Department of Psychiatry, Temerty Faculty of Medicine, University of Toronto, Toronto, Canada

**\*For correspondence:**
Luisa.Fassi@mrc-cbu.cam.ac.uk
(LF);
r.cohenkadosh@surrey.ac.uk
(RCK)

[†]These authors contributed equally to this work

**Competing interest:** The authors declare that no competing interests exist.

**Abstract** In recent years, there has been debate about the effectiveness of treatments from different fields, such as neurostimulation, neurofeedback, brain training, and pharmacotherapy. This debate has been fuelled by contradictory and nuanced experimental findings. Notably, the effectiveness of a given treatment is commonly evaluated by comparing the effect of the active treatment versus the placebo on human health and/or behaviour. However, this approach neglects the individual's subjective experience of the type of treatment she or he received in establishing treatment efficacy. Here, we show that individual differences in *subjective treatment* - the thought of receiving the active or placebo condition during an experiment - can explain variability in outcomes better than the actual treatment. We analysed four independent datasets (N = 387 participants), including clinical patients and healthy adults from different age groups who were exposed to different neurostimulation treatments (transcranial magnetic stimulation: Studies 1 and 2; transcranial direct current stimulation: Studies 3 and 4). Our findings show that the inclusion of *subjective treatment* can provide a better model fit either alone or in interaction with *objective treatment* (defined as the condition to which participants are assigned in the experiment). These results demonstrate the significant contribution of subjective experience in explaining the variability of clinical, cognitive, and behavioural outcomes. We advocate for existing and future studies in clinical and non-clinical research to start accounting for participants' subjective beliefs and their interplay with objective treatment when assessing the efficacy of treatments. This approach will be crucial in providing a more accurate estimation of the treatment effect and its source, allowing the development of effective and reproducible interventions.

## eLife assessment

This is an **important** report that has implications for both the brain stimulation field and beyond. The strength of evidence provided is quite **convincing**. The major strength of this work is the recognise the importance of participant expectation in brain stimulation studies.

**eLife digest** Neuromodulation is a type of intervention that relies on various non-invasive techniques to temporarily stimulate the brain and nervous system. It can be used for the treatment of depression or other medical conditions, as well as the improvement of cognitive abilities such as attention. However, there is conflicting evidence regarding whether this approach has beneficial effects.

Most studies aiming to assess the efficiency of a treatment rely on examining the outcomes of people who received the intervention in comparison to participants who undergo a similar procedure with no therapeutic effect (or placebo). However, the influence of other, 'subjective' factors on these results – such as the type of intervention participants think they have received – remains poorly investigated.

To bridge this gap, Fassi and Hochman et al. used statistical modeling to assess how patients' beliefs about their treatment affected the results of four neuromodulation studies on mind wandering, depression and attention deficit hyperactivity disorder symptoms. In two studies, participants' perceptions of their treatment status were more strongly linked to changes in depression scores and mind-wandering than the actual treatment. Results were more nuanced in the other two studies. In one of them, participants who received the real neuromodulation but believed they received the placebo showed the most improvement in depressive symptoms; in the other study, subjective beliefs and objective treatment both explained changes in inattention symptoms.

Taken together, the results by Fassi and Hochman et al. suggest that factoring in patients' subjective beliefs about their treatment may be necessary in studies of neuromodulation and other interventions like virtual reality or neurofeedback, where participants are immersed in cutting-edge research settings and might therefore be more susceptible to develop beliefs about treatment efficacy.

## Introduction

A substantial amount of research from medicine, neuroscience, psychology, and education aims to establish the effectiveness of different treatments, such as drugs, cognitive training, biofeedback, and neurostimulation, in both clinical and non-clinical populations. However, the research findings from these fields tend to be heterogeneous. As a result, there has been increased scepticism among researchers about the efficacy of these treatments (*Lampit et al., 2014*; *López-Alonso et al., 2014*; *Sitaram et al., 2017*).

In recent years, neuromodulation has been studied as one of the most promising treatment methods (*De Ridder et al., 2021*). Further, one particular form of neuromodulation, transcranial magnetic stimulation (TMS), has been approved by regulatory bodies in multiple countries, including the US Food and Drug Administration (FDA), and is used as an evidence-based treatment for patients with migraine, major depression, obsessive-compulsive disorder, and smoking addiction (*Hallett, 2007*; *Walsh and Cowey, 2000*). Moreover, TMS and other neuromodulatory devices, such as transcranial-focused ultrasound and electrical stimulation (tES), have been highlighted as a potential treatment for psychiatric, neurological, and neurodevelopmental disorders (*Grover et al., 2021*; *Khedr et al., 2005*; *McGough et al., 2019*), and they have also been used to enhance various mental processes, including attention, memory, language, mathematics, and intelligence in healthy populations (*Santarnecchi et al., 2015*). These encouraging findings have raised hope for the potential application of these techniques within and outside the clinic (*Dubljević et al., 2014*).

Despite some encouraging results on the beneficial effects of both TMS and tES, contradictory findings have emerged across different studies (*Horvath et al., 2015*; *Medina and Cason, 2017*; *Parkin et al., 2015*; *Wang et al., 2018*; *Westwood et al., 2017*). Several factors have been pointed at as plausible reasons for the heterogeneity in research results (*Filmer et al., 2020*; *Guerra et al., 2020*; *van Bueren et al., 2021*). However, a crucial factor that researchers have largely overlooked is the extent to which subjective beliefs can explain variability in treatment efficacy. Here, we address this gap by examining whether modelling participants' beliefs about receiving the placebo or active treatment can account for changes in clinical, cognitive and behavioural outcomes.

Participants who take part in TMS and tES studies consistently report various perceptual sensations, such as audible clicks, visual disturbances, and cutaneous sensations (*Davis et al., 2013*).

Consequently, they can discern when they have received the active treatment, making subjective beliefs and demand characteristics potentially influencing performance (*Polanía et al., 2018*). To account for such non-specific effects, sham (placebo) protocols have been employed. For transcranial direct current stimulation (tDCS), the most common form of tES, various sham protocols exist. A review by *Fonteneau et al., 2019* shows that 84% of 173 studies used similar sham approaches to an early method by *Gandiga et al., 2006*. This initial protocol had a 10 s ramp-up followed by 30 s of active stimulation at 1 mA before cessation, differently from active stimulation that typically lasts up to 20 min. However, this has been adapted in terms of intensity and duration of current, ramp-in/-out phases, and the number of ramps during stimulation. Similarly, in sham TMS, the TMS coil may be tilted or replaced with purpose-built sham coils equipped with magnetic shields, which produce auditory effects but ensure no brain stimulation (*Duecker and Sack, 2015*). By using surface electrodes, the somatosensory effects of actual TMS can also be mimicked. Overall, these types of sham stimulation aim to simulate the perceptual sensations associated with active stimulation without substantially affecting cortical excitability (*Fritsch et al., 2010*; *Nitsche and Paulus, 2000*). As a result, sham treatments should allow controlling for participants' specific beliefs about the type of stimulation received.

Previous studies have addressed whether manipulating participants' expectations about the effects of either active or sham stimulation can moderate treatment efficacy (*Haikalis et al., 2023*; *Rabipour et al., 2018*). However, to our knowledge, these studies have not examined whether individual differences in participants' subjective experience of receiving the active or sham treatment provide a better model fit than the condition to which participants are assigned in the study. We term the former *subjective treatment* and the latter *objective treatment*.

The above consideration becomes particularly crucial when considering that the experimental design of most randomised controlled trials (RCTs) involves recording whether participants believed they received the active or placebo treatment. While it is common practice to assess experimental blinding using this data, the explanatory power of individual differences in *subjective treatment* is rarely, if at all, considered. This is based on the assumption that if no differences emerged at the group level in participants' guess for receiving the active vs the placebo treatment (i.e. if experimental blinding was successful), placebo effects could not explain the obtained results.

Here, we hypothesise that such an assumption can be erroneous and aim to explore how accounting for differences in subjective beliefs can shed light on the conclusions of previous treatment studies. Moreover, we introduce a simple and straightforward approach that could be used to analyse existing data and guide future clinical and fundamental research to examine whether *subjective treatment* explains variability in experimental outcomes over and beyond *objective treatment*. Below, we demonstrate this approach by reanalysing four independently published neurostimulation studies (including TMS and tES) that test clinical and non-clinical samples from different age groups (*Blumberger et al., 2016*; *Filmer et al., 2019*; *Leffa et al., 2022*; *Kaster et al., 2018*). The data and the codebook of the analyses are available on the OSF (https://osf.io/rztxu/).

## Results
### Study 1
Repetitive TMS (rTMS) is a method for treating depression that has been approved by the FDA (*Connolly et al., 2012*). In study 1 (*Blumberger et al., 2016*), patients aged 18–85 with treatment-resistant depression (N = 121) were randomised to receive either bilateral rTMS, unilateral left-rTMS, or sham rTMS for 3 or 6 weeks (*objective treatment*). We examined whether participants' beliefs about receiving active or sham stimulation (*subjective treatment*) explained changes in depression over time. In this study, *subjective treatment* was based on participants' reports of whether they thought they received active or sham rTMS, inquired at the end of treatment (i.e., week 6). Further details on participant groupings based on the *objective treatment* and *subjective treatment* can be found in the codebook, separately for each study, and figure supplements (e.g. *Figure 1—source data 1*).

A linear mixed model with depression scores, measured by the Hamilton Depression Rating scale (HAMD-17), was fitted to the data for weeks 1–6. The baseline model included *time* (week 1/week 3/ week 6) as a main effect, as well as the interaction of *time* and *objective treatment*. We first added to this model *subjective treatment* as a main effect. Next, we extended the model to include the two-way interaction of *time* and *subjective treatment*. Lastly, we considered a model with the three-way

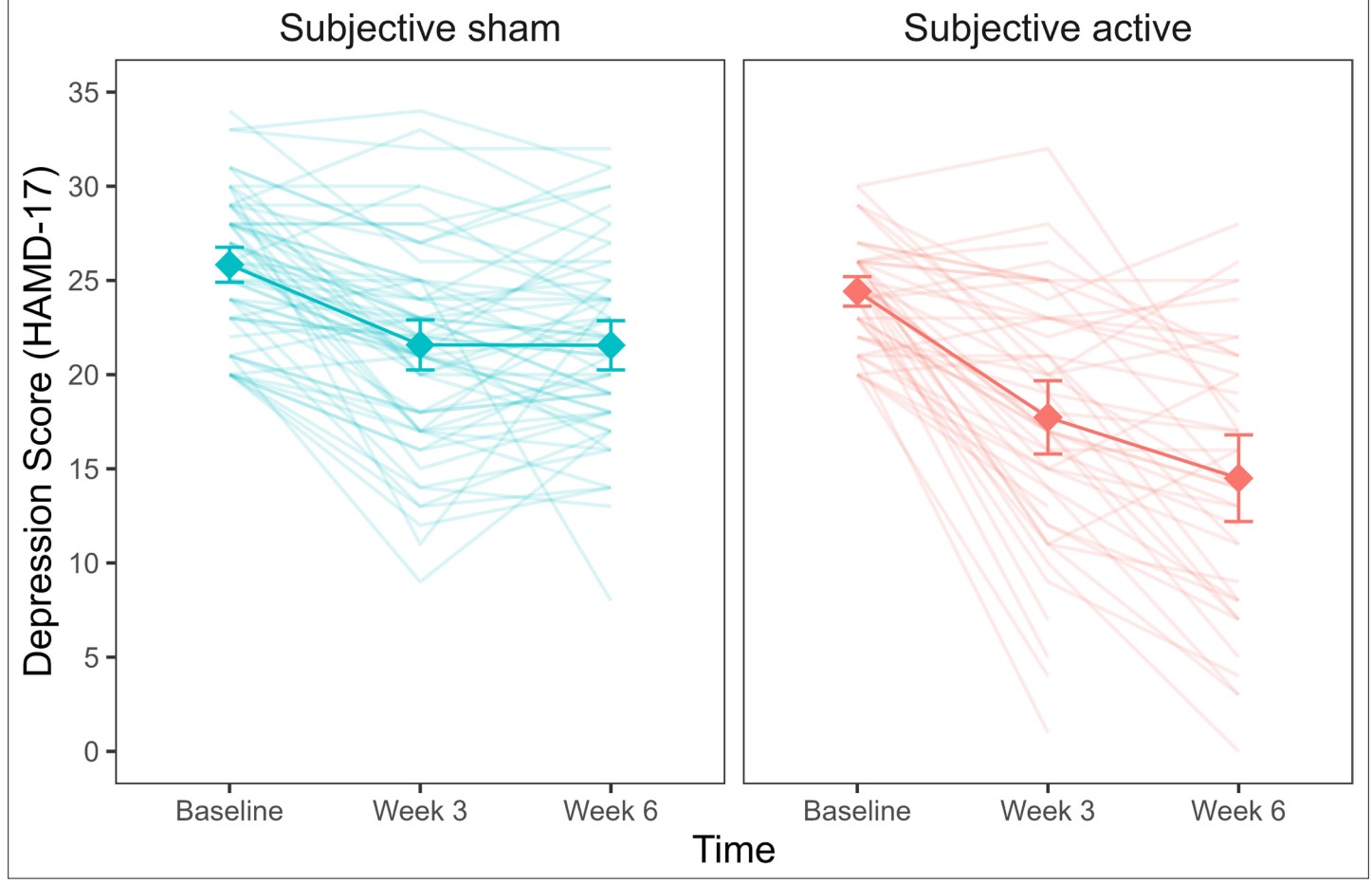

**Figure 1.** Depression scores as a function of subjective treatment over time. Each diamond represents the mean depression score (HAMD-17) for the time points (baseline, week 3, week 6), and each line in the background represents a patient. Error bars represent ±1 standard error of the mean.

The online version of this article includes the following source data for figure 1:

**Source data 1.** Table with sample size (n) for objective and subjective treatment.

**Source data 2.** Table with summary statistics for the model comparison with HAMD-17 depressive symptoms as outcome.

**Source data 3.** Table with summary statistics for the model comparison with HAMD-17 depressive symptoms as outcome.

interaction of *time*, *subjective treatment,* and *objective treatment*. Our results showed that the two-way interaction between *subjective treatment* and *time* led to a significantly higher model fit (Bayesian Information Criterion (*BIC*) = 2027.48, Akaike Information Criterion (*AIC*) = 1985.62, *p*<0.001; see *Figure 1—source data 2*). Hence, our analysis suggests that participants' subjective experience about the treatment accounted for variability in depression scores over time, while the actual treatment condition to which participants were assigned did not. As shown in *Figure 1*, participants who thought they received active stimulation showed a steeper decrease in depression over time than participants who thought they received sham. The interaction of subjective treatment and time was significant in weeks 3 and 6. We used contrasts to break down this interaction and compare depression scores at weeks 3 and 6 to depression at baseline (week 0) between participants who reported active vs sham as *subjective treatment*. Our results showed that depression scores were lower for participants who thought they were receiving active compared to sham stimulation at both weeks 3 (*b* = -3.15, *t*(321) = -3.43, *p*< 0.001) and 6 (*b* **=** -6.72, *t*(321) = -6.84, *p*< 0. 001).

We next examined whether variability in depression scores was explained by both *objective* and *subjective treatment*. To this aim, we run a model comparison adding *objective treatment* first and, secondly, the interaction of *objective treatment* with *time* to a baseline model already including the interaction of *subjective treatment* and *time*. Our results showed that the inclusion of neither *objective treatment* nor the *objective treatment* by *time* interaction led to a better model fit (*Figure 1—source*

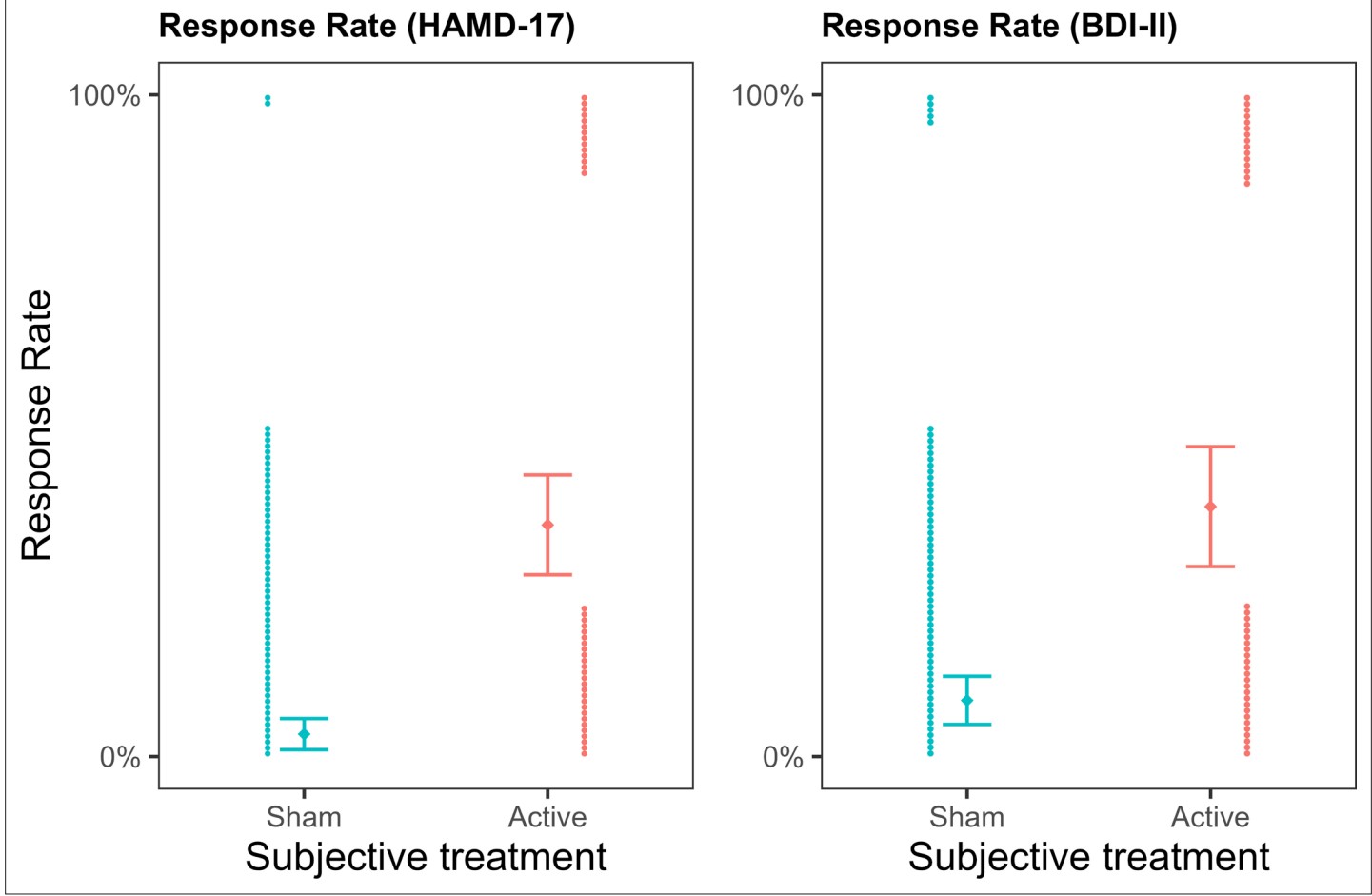

**Figure 2.** Depression response rates as a function of subjective treatment. The left plot presents the contribution of subjective treatment on the response rate of the Hamilton Depression Rating Scale (HAMD-17), and the right plot presents the contribution of subjective treatment on the Beck Depression Inventory II (BDI-II). Each dot represents an individual patient, stacked towards 100% representing a response or 0% representing no response. Error bars represent ±1 standard error of the mean.

The online version of this article includes the following source data for figure 2:

**Source data 1.** Table with summary statistics for the model comparison with HAMD response rates as outcome.

**Source data 2.** Table with summary statistics for the model comparison with HAMD response rates as outcome.

**Source data 3.** Table with summary statistics for the model comparison with BDI-II response rates as outcome.

**Source data 4.** Table with summary statistics for the model comparison with BDI-II response rates as outcome.

*data 3*). Therefore, a statistical model that includes the participants' subjective experience of receiving the real or sham treatment at baseline fits the observed data better than a statistical model that only includes the actual treatment allocation.

We also investigated whether *subjective treatment* could explain variability in participants' response and remission rates. In the study, the response rate was defined as a >50% reduction in depressive symptomatology and was binary coded. A mixed binomial model with HAMD-17 response rate as the outcome was fitted to the data. The baseline model included only *objective treatment* as a predictor and was compared to an updated model, including *subjective treatment* as a main effect. Given that response rates were measured only once, time did not vary and was therefore omitted from the model. We compared the model with *subjective treatment* as a main effect to the model including the *subjective treatment* by *objective treatment* interaction. Our results showed that the model with *subjective treatment* as a main effect led to a significantly better fit (*deviance* = 15.08, *p* = 0.0001; see *Figure 2—source data 1*). As shown in *Figure 2*, response rates were higher for participants who reported thinking they received the active compared to the sham treatment (*log(OR)* = 2.61, *z* = 3.21,

*SE* = 0.81, *p* = 0.001). On the contrary, when we examined whether the addition of *objective treatment* to a model already including *subjective treatment* led to a better fit, this was not the case (*deviance* = 1.40, *p* = 0.496; see *Figure 2—source data 2*). Therefore, treatment allocation did not explain changes in patients' depression when subjective beliefs were already accounted for in the model.

The same pattern of results was replicated for response rates calculated based on another depression scale, the Beck Depression Inventory II (BDI-II), where *subjective treatment* as a main effect led to a significantly better model fit (*deviance* = 10.81, *p* = 0.001; see *Figure 2—source data 3*), and participants who reported the active subjective treatment showed higher response rates (*log(OR)* = 1.85, *z* = 3.06, *SE* = 0.60, *p* = 0.002). In contrast, *objective treatment* did not provide a better model fit than *subjective treatment* (*deviance* = 0.27, *p* = 0.873; see *Figure 2—source data 4*).

Additionally, in the study, participants were classified as either remitters or non-remitters based on blinded clinical ratings at the end of weeks 3 and 6, defined by a HAMD-17 score ≤7. We conducted a survival analysis to examine whether subjective treatment explained variability in remission rates. The results supported the idea that patients who reported they subjectively believed receiving active stimulation showed higher remission rates than patients who believed they received sham. We found that, for *objective treatment*, the survival curves did not significantly differ between the active and sham condition (*Gehan–Breslow–Wilcoxon test(1)* = 3.72, *p* = 0.053), indicating that remission rates did not differ for patients who received active rTMS compared to patients who received sham. On the contrary, for subjective treatment, a significant difference emerged (*Gehan–Breslow–Wilcoxon test(1)* = 18.16, *p*< 0.001). Specifically, patients who reported they believed receiving active stimulation showed higher remission rates than patients who believed they received sham (*Gehan–Breslow–Wilcoxon test(1)* = 5.12, *p* = 0.020).

## Study 2

In study 2 (*Kaster et al., 2018*), 52 participants aged 60-85 diagnosed with late-life depression were randomised to active or sham high-dose deep rTMS. Compared to standard rTMS, deep rTMS with the H1 coil has been designed to stimulate deeper and larger areas of the cortex (primarily the left dorsolateral prefrontal cortex - DLPFC - and portions of the right DLPFC). We examined whether, also in this case, *subjective treatment* accounted for changes in participants' depression scores, despite the use of a different sample and TMS technique. Notably, in contrast to the other studies, participants were asked to report whether they thought they received the active or sham treatment after the first week of treatment rather than at the end (i.e., fourth week). This avoids that *subjective treatment* - as inquired at the end of the study - would inherently be biased due to the clinical change the patient experienced as a result of the intervention. The breakdown of participants in respect to *objective treatment* and *subjective treatment* is reported in *Figure 3—source data 1*.

A linear mixed model with Hamilton Depression Rating Scale (HDRS-24) score as the outcome was fitted to the data for weeks 1–4. As in study 1, we first compared the baseline model, including *time* and its interaction with the *objective treatment,* to a model including the interaction of *subjective treatment* by *time*. The latter model was then compared to a three-way interaction model with *subjective treatment* by *objective treatment* by *time*. Our results showed that the three-way interaction model led to a significantly better fit (*AIC* = 1601.91, *BIC* = 1668.75, *p* = 0.011; see *Figure 3—source data 2 and 3*). Hence, participants' beliefs explained variability in depression scores over time in relation to the experimental allocation. To examine contrasts of the three-way interaction, we analysed the differences between the objective and subjective treatment each week compared to the baseline (*Figure 3*). We found a steeper decrease in depression from baseline to week 3 (*b* = 8.79, *t*(102.57) = 2.01, *SE* = 4.37, *p* = 0.047) and from baseline to week 4 (*b* = 9.19, *t*(103.80) = 2.10, *SE* = 4.39, *p* = 0.039). In both cases, the scores for active *objective treatment* and active *subjective treatment* were higher than the sham treatment. Another way to explore the three-way interaction is by investigating the polynomial contrasts of the time variable between the objective and subjective treatment conditions. The analysis showed that the objective and subjective treatment differed in the linear contrast of time (*b* = 28.03, *t*(182.30) = 3.13, *p* = 0.002). The contrast showed a negative slope throughout the weeks that were significantly different between the objective treatment levels in the subjective sham treatment (*b* = -19.63, *t*(181.05) = -2.77, *p* = 0.001), but not for the subjective active treatment (*b* = 8.40, *t*(184.33) = 1.52, *p* = 0.128). Thus, the results show that the steepest change in depression occurred among those who received the

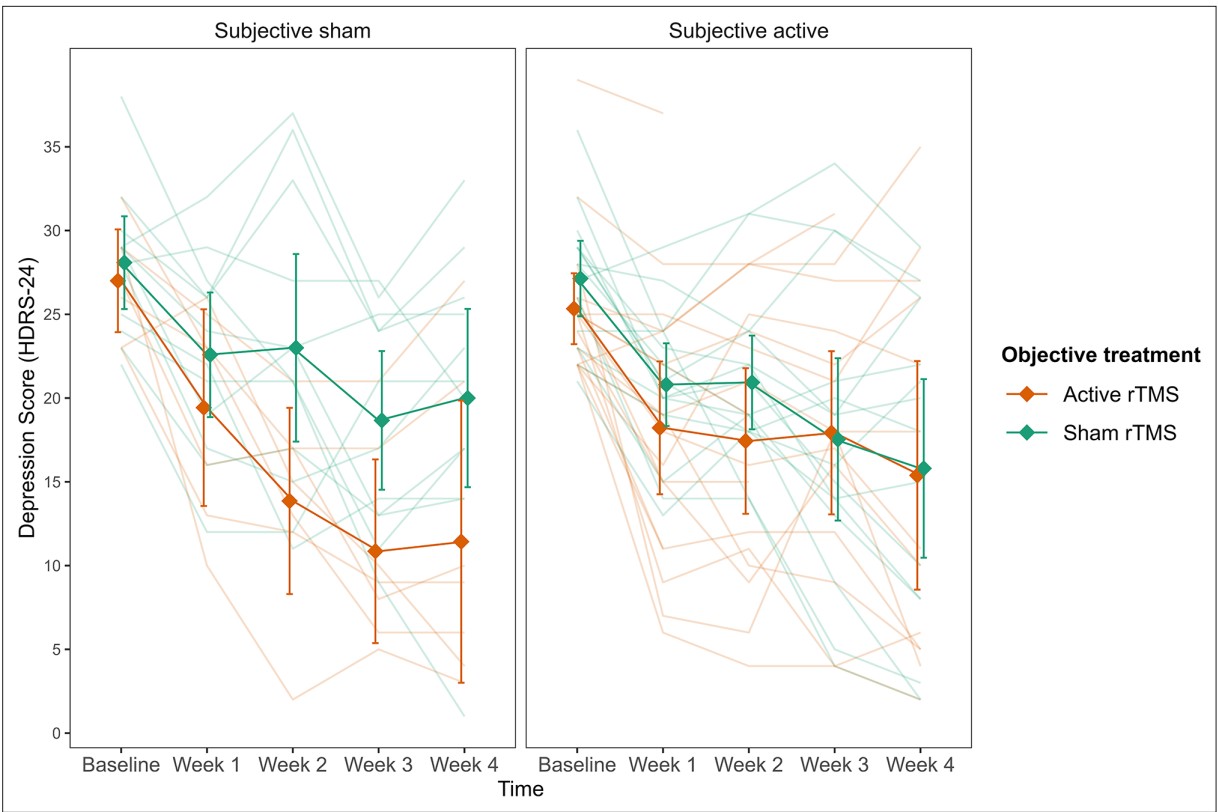

**Figure 3.** Depression scores as a function of the three-way interaction between subjective treatment, objective treatment, and time. Subjective sham treatment drives the difference between objective treatments in depression scores. The left plot shows subjective sham treatment, and the right plot shows subjective active treatment. Each line in the background represents a patient. Error bars represent ±1 standard error of the mean.

The online version of this article includes the following source data for figure 3:

**Source data 1.** Table with sample size (n) for objective and subjective treatment.

**Source data 2.** Table with summary statistics for the model comparison with HDRS-24depressive symptoms as outcome.

**Source data 3.** Table with summary statistics for the model comparison with HDRS-24depressive symptoms as outcome.

active treatment but believed they received the sham treatment (compared to those who believed they received the active treatment).

We further investigated whether *subjective treatment* could provide a better model fit for the patients' remission and response rates than *objective treatment*. To this aim, we fitted two mixed binomial models with remission and response rates as the outcomes. Time was not considered in this case because both remission and response rates were collected only once at the end of the fourth week. In line with our previous results, we found that the interaction of *subjective treatment* by *objective treatment* was significantly better at predicting remission rates (*deviance* = 4.47, *p* = 0.035; see *Figure 4—source data 1 and 2*) and response rates (*deviance* = 8.17, *p*=0.004; see *Figure 4—source data 3 and 4*). For remission rates, we found a significant two-way interaction between *objective treatment* and *subjective treatment* (*log(OR)* = 0.81, *z* = 1.99, *SE* = 0.41, *p* = 0.047, *Figure 4*). While the effect did not differ significantly between active and sham rTMS as the *objective treatment* when participants thought they received the active stimulation (*b* = -1.01, *z* = -0.82, *SE* = 1.23, *p* = 0.410), higher remission rates were found when patients thought they received sham (*b* = 2.69, *SE* = 1.29, *z* = 2.08, *p* = 0.038). These results were replicated when we considered response rates as the outcome (*Figures 4*) for which we found a significant two-way interaction (*log(OR)* = 1.02, *z* = 2.58, *SE* = 0.4, *p* = 0.010). Again, for participants who thought they received the active stimulation, remission did not differ significantly between active and sham rTMS as the *objective treatment* (*b* = -1.7, *z* = -1.44, *SE* = 1.18, *p* = 0.150). On the contrary, when participants thought they received sham stimulation, they showed higher response rates in the active compared to sham rTMS as the *objective treatment* (*b* = 2.38, *SE* = 1.05, *z* = 2.26, *p* = 0.020).

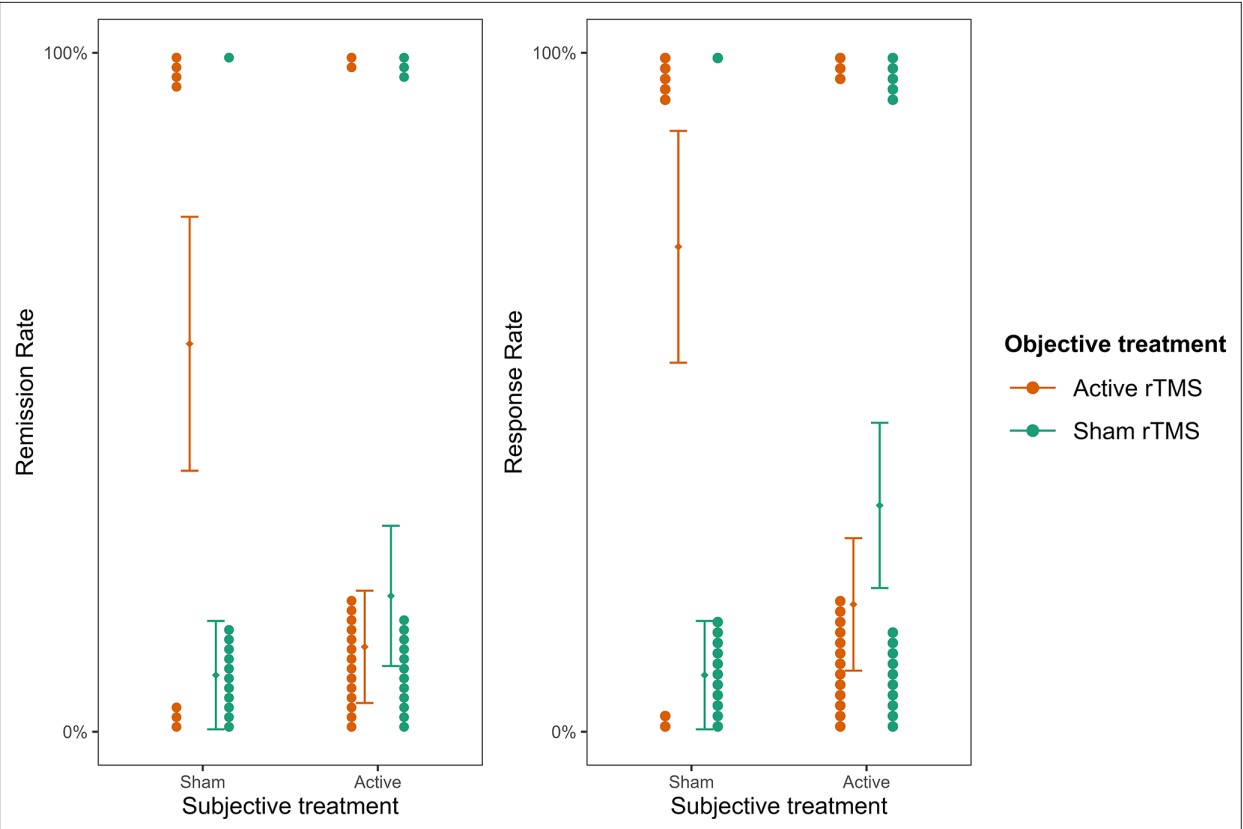

**Figure 4.** Remission and response rates as a function of subjective and objective treatment. The plots present the contribution of subjective and objective treatment on the HDRS-24 remission and response rates. Within each plot, the left columns present the contribution of objective active treatment and the right columns the contribution of objective sham treatment, separately for the two levels of subjective treatment. Each dot represents an individual patient and is stacked towards 100% representing a response or 0% representing no response. Error bars represent ±1 standard error of the mean.

The online version of this article includes the following source data for figure 4:

**Source data 1.** Table with summary statistics for the model comparison with HDRS-24 remission rates as outcome.

**Source data 2.** Table with summary statistics for the model comparison with HDRS-24 remission rates as outcome.

**Source data 3.** Table with summary statistics for the model comparison with HDRS-24response rates as outcome.

**Source data 4.** Table with summary statistics for the model comparison with HDRS-24response rates as outcome.

## Study 3

In study 3, the researchers examined the effect of home-based tDCS treatment used for 4 weeks on a clinical group of adults diagnosed with ADHD (*Leffa et al., 2022*; N = 64). The primary outcome measure was symptoms of inattention taken from a clinician-administered questionnaire (Adult ADHD Self-report Scale; CASRS-I). Data on participants' beliefs reflecting *subjective treatment* was collected at the end of the experiment. The breakdown of participants by *objective treatment* and *subjective treatment* in the sample can be found in *Figure 5—source data 1*.

In line with the studies above, we first investigated the addition of *subjective treatment* to a model accounting for *objective treatment* between the baseline and the last assessments. Including *subjective treatment* led to a better model fit (*AIC* = 593.81, *BIC* = 609.79, *p*< 0.001; see *Figure 5—source data 2*). Subsequent contrast analysis revealed that inattention scores for participants who believed they were getting the active treatment were significantly lower compared to those who believed they belonged to the sham group *(b = −3.33, t(100)* = -3.35, *SE* = 0.99, *p* = 0.001, see *Figure 5*). This finding provides further evidence supporting the contribution of *subjective treatment* over *objective treatment*, extending our previous results to another mental health condition, population, and tES method.

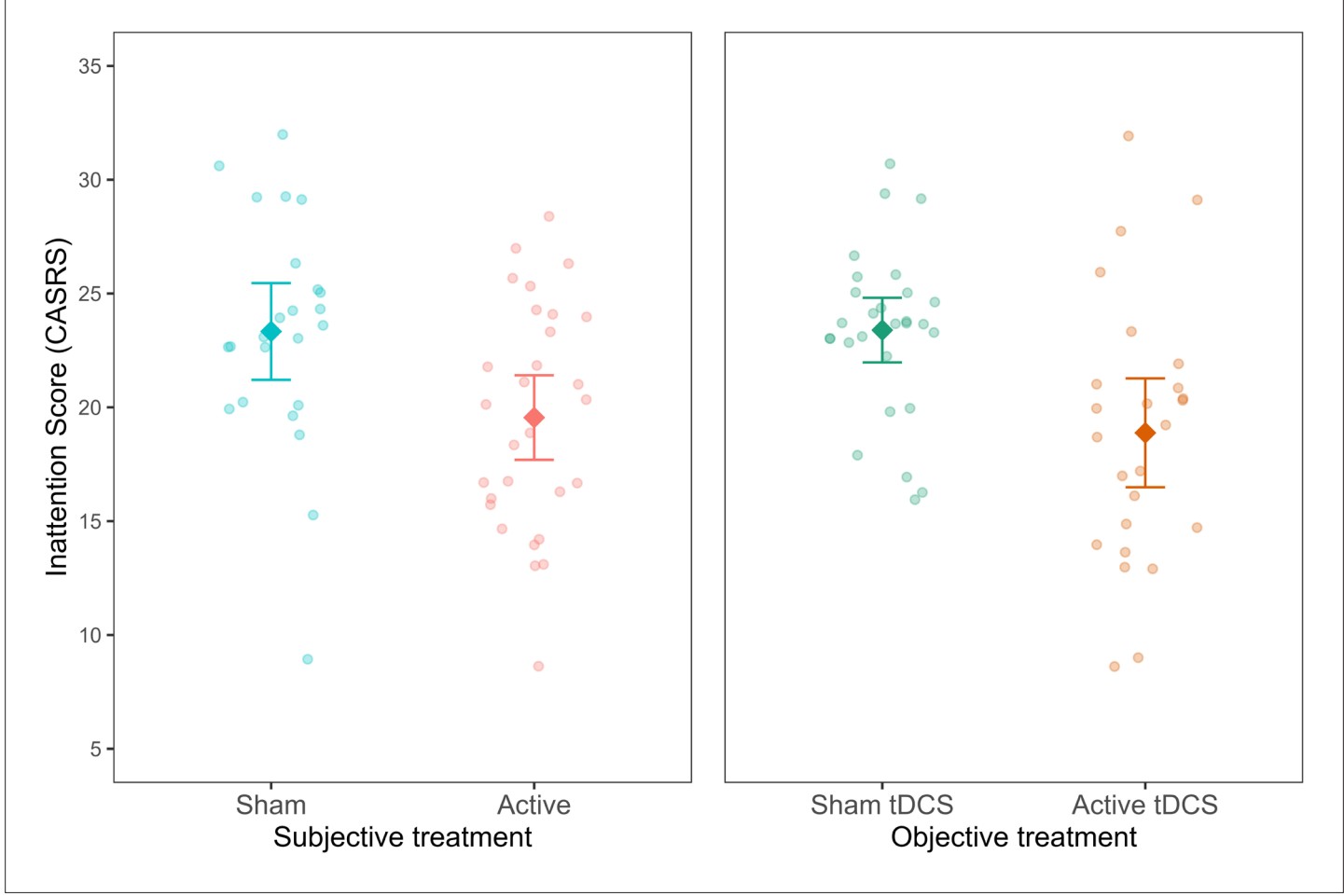

**Figure 5.** Inattention symptoms as a function of subjective and objective treatment. The left plot shows the contribution of subjective treatment, and the right plot shows the contribution of objective treatment. Each dot represents an individual patient. Error bars represent ±1 standard error of the mean.

The online version of this article includes the following source data for figure 5:

**Source data 1.** Table with sample size (n) for objective and subjective treatment.

**Source data 2.** Table with summary statistics for the model comparison with CASRSinattention symptoms as outcome.

**Source data 3.** Table with summary statistics for the model comparison with CASRSinattention symptoms as outcome.

Next, we investigated whether a model including *objective treatment* could explain variability in a model already including *subjective treatment*. Differently from studies 1 and 2, where this addition was not found significant, here, the addition of the *objective treatment* was significant (*AIC* = 593.81, *BIC* = 609.79, *p*< 0.001; see *Figure 5—source data 3*). As expected, the contrast showed lower inattention symptoms in the objective active treatment group *(b* = -4.17, *t*(100) = −4.21, *SE* = 0.99, *p*< 0.001). Thus, *subjective treatment* did not fully overrule the contribution of objective treatment to research outcomes. As later expanded on, this finding demonstrates the varied explanatory power that *subjective treatment* can have in relation to various types of tES treatments.

## Study 4

In study 4, we extended our results beyond clinical populations by examining the effects of different doses (current intensity) of tDCS on mind-wandering in healthy participants (N = 150; *Filmer et al., 2019*). Similar to studies 1 and 3, participants were asked about subjective beliefs at the end of the experiment. For this study, we tested whether not only *subjective treatment* but also *subjective dosage* (participants' beliefs of the strength of the stimulation they received) could explain variability

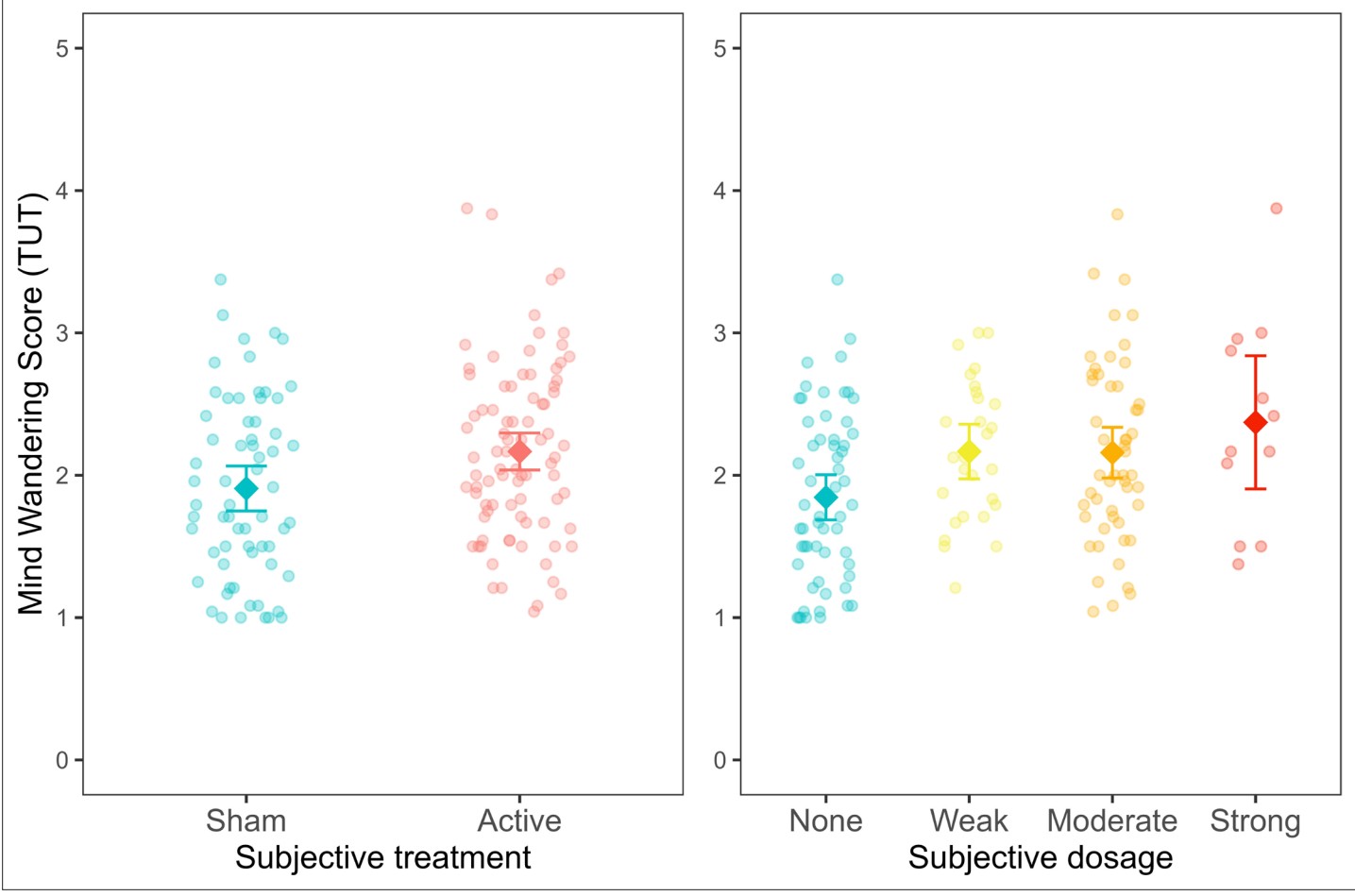

**Figure 6.** Mind wandering scores, based on the task-unrelated thought (TUT) average across experimental trials, as a function of subjective treatment and subjective dosage. Each dot represents a participant. Error bars represent ±1 standard error of the mean.

The online version of this article includes the following source data for figure 6:

**Source data 1.** Table with sample size (n) for objective and subjective treatment.

**Source data 2.** Table with summary statistics for the model comparison with mind wandering scores as outcome.

**Source data 3.** Table with summary statistics for the model comparison with mind wandering scores as outcome.

**Source data 4.** Table with summary statistics for the model comparison with mind wandering scores as outcome.

**Source data 5.** Table with summary statistics for the model comparison with mind wandering scores as outcome.

in the results attributed originally to *objective treatment*. The breakdown of participants to *objective treatment* and *subjective treatment* in the sample can be found in *Figure 6—source data 1*.

A linear regression model with average mind wandering scores calculated over the whole experimental session was fitted to the data. In line with studies 1 and 3, *subjective treatment* contributed to a significantly better model fit. Specifically, participants' beliefs explained variability in mind wandering when *subjective treatment* was included as a main effect on top of *objective treatment* (AIC = 284.72, BIC = 305.80, *p* = 0.045; see *Figure 6—source data 2*). Furthermore, as shown in *Figure 6*, participants who believed they received active treatment showed higher mind-wandering levels than those who reported they believed to receive sham treatment (*b* = −0.11, *SE* = 0.05, *t*(144) = -2.01, *p* = 0.046).

The experimental design in this study allowed us also to expand our previous findings by examining the contribution of *subjective dosage* to a model including *objective treatment* (AIC = 282.90, BIC = 310, *p* = 0.025; *Figure 6—source data 4*). In this regard, we found that mind wandering increased for people reporting weak (*b* = −0.31, *SE* = 0.14, *t*(142) = -2.23, *p* = 0.03), moderate (*b* = −0.27, *SE* = 0.12, *t*(142) = -2.24, *p* = 0.027) and strong (*b* = 0.47, *SE* = 0.19, *t*(142) = 2.41, *p* = 0.017)

*subjective dosage* compared to none. These results indicate that mind wandering increased proportionally as the subjective dosage increased (from none to strong). Conversely, our results showed that participants' *objective treatment* did not lead to a better model fit neither when added to a model including *subjective treatment* (AIC = 284.72, BIC = 305.8, p = 0.093; see *Figure 6—source data 3*) nor when added to a model including *subjective dosage* (AIC = 282.9, BIC = 310.0, p = 0.106; see *Figure 6—source data 5*). These findings highlight that participants' beliefs regarding the type of treatment received and their subjective experience of the treatment dosage can explain variability in cognitive performance.

## Discussion

In this work, we used a novel approach to examine whether and to what extent participants' subjective beliefs may account for variability in research outcomes. To this aim, we analysed four independent datasets from the field of neurostimulation; specifically, two rTMS RCTs in patients with depression (*Blumberger et al., 2016*; *Kaster et al., 2018*), one tDCS study in adults with ADHD (*Leffa et al., 2022*), and another tDCS study in a healthy adult sample (*Filmer et al., 2019*).

We demonstrate that participants' subjective beliefs about receiving the active vs control (sham) treatment are an important factor that can explain variability in the primary outcome and, in some cases, fit the observed data better than the actual treatment participants received during the experiment. Specifically, in studies 1 and 4, the fact that participants thought to be in the active or control condition explained variability in clinical and cognitive scores to a more considerable extent than the *objective treatment* alone. Notably, the same pattern of results emerged when we replaced *subjective treatment* with *subjective dosage* in the fourth experiment, showing that subjective beliefs about treatment intensity also explained variability in research results better than objective treatment. In contrast to studies 1 and 4, studies 2 and 3 showed a more complex pattern of results. Specifically, in study 2 we observed an interaction effect, whereby the greatest improvement in depressive symptoms was observed in the group that received the active objective treatment but believed they received sham. Differently, in study 3, the inclusion of both subjective and objective treatment as main effects explained variability in symptoms of inattention. Overall, these findings suggest the complex interplay of objective and subjective treatment. The variability in the observed results could be explained by factors such as participants' personality, disorder type and severity, prior treatments, knowledge base, experimental procedures, and beliefs of the research team, all of which could be interesting avenues for future studies to explore.

An important question arising from our findings relates to the causal role of subjective beliefs. This question is a complex one to answer and falls outside the scope of this study. Based on the goal of testing blinding efficacy, it is a standard practice for current treatment studies to record data on subjective beliefs only at the end of the experiment rather than before and throughout. This was the case in studies 1, 3, and 4. While in study 3, both *objective* and *subjective treatment* explained unique variance in the clinical outcomes, in studies 1 and 4 it was impossible to conclude whether participants' beliefs, which were captured by the *subjective treatment*, affected experimental outcomes or, on the contrary, whether participants' changes in performance or symptoms throughout the studies influenced beliefs regarding treatment allocation. Therefore, it is important to consider that *subjective treatment* could also capture the changes faced by participants in placebo-controlled trials, whereby, if they feel better, it would be hard psychologically to report that they believe it was due to the placebo. Notably, study 2 used a less common approach, in which subjective beliefs about treatment allocation were queried after the first week of the study. In this case, significant results emerged only 2 weeks after this inquiry (i.e., weeks 3 and 4). Given that subjective beliefs about treatment allocation were documented before the emergence of changes in any clinical outcome, it is more plausible that participants' beliefs would have affected experimental outcomes rather than vice versa.

Based on the above considerations, future studies should strive to record data on subjective beliefs at different time points: before, during, and after the experiment. This will allow mapping the way subjective beliefs might be differentially associated with experimental results depending on the considered study and treatment type. However, we acknowledge two caveats of this suggestion. Firstly, participants may be more prone to pay attention to their treatment allocation and, consequently, figure out their assigned condition. Secondly, recording subjective beliefs at multiple time points might interfere with the effects of the treatment. For instance, patients might suppress their

response for the fear that the treatment received is a placebo (*Sonawalla and Rosenbaum, 2002*). An alternative approach could entail deception, whereby all participants are told they received the active treatment. While this raises an ethical concern, such an approach would: (1) allow minimising the effect of subjective beliefs on research outcomes, and (2) hold more ecological validity, as it would mimic the way approved treatments are delivered in the clinic, where all patients know to be receiving the active treatment (*Fonteneau et al., 2019*).

While this study focuses only on neuromodulation techniques, we want to highlight that the proposed approach can be applied to other forms of treatment (e.g. pharmacological studies, cognitive training) tested as part of standard experiments or RCTs. It is worth noting that the contribution of subjective beliefs to experimental results might be even more enhanced when considering interventions carried out in seemly cutting-edge research settings, such as experiments involving virtual reality, neurofeedback paradigms, and other types of brain–computer interfaces. In such cases, participants might be more susceptible to forming specific expectations about treatment effects (*Fonteneau et al., 2019*; *Thibault et al., 2017*; *Mikellides et al., 2022*). Therefore, the explanatory power of subjective beliefs could be intensified compared to more traditional forms of treatment, such as pharmacology.

One question emerging from this study is whether these results, observed with self-report measures, would apply to more objective behavioural outcomes (e.g. sensorimotor recovery in stroke patients, improvements in fluid intelligence in participants with learning difficulties) and neural functions (e.g. functional connectivity). We argue that the answer to this question is likely to be positive since placebo effects have been shown to impact not only behaviour but also brain activity (*Fonteneau et al., 2019*; *Hashmi, 2018*; *Oken et al., 2008*; *Schmidt et al., 2014*). However, independent of this possibility, the contribution of *subjective treatment* to explaining variability in self-reported outcomes should not be underestimated. Noteworthy, in most RCTs investigating the effect of different treatments on clinical and subclinical groups (e.g. depression, chronic pain, eating disorders, attention-deficit/hyperactivity disorder), some of which have also been approved by the FDA, the measurement of symptomatology is mostly based on self-reported outcomes, such as questionnaires. This consideration makes the case of *subjective treatment* even stronger, hinting at the potential role of this factor in explaining experimental results across a variety of experimental outcomes and treatment types.

While our study examined the explanatory power of subjective beliefs about receiving treatment, neither of the four studies (similar to most studies in the field) collected data on participants' expectations. Indeed, as recently shown by *Parong et al., 2022*, expectations regarding the effectiveness of cognitive training (i.e. whether it will increase or decrease performance) can significantly modulate its effect. Thus, investigating the interplay between expectations and *subjective treatment* could allow examining the directionality and strength of the effect of *subjective treatment* on the outcome of interest. For instance, some participants may expect a treatment to improve their capabilities or symptoms. In contrast, others could expect even the opposite, and the level of these expectations can vary during the intervention. These factors could, in turn, impact individual variability in *subjective treatment*. Arguably, when questioned early, *subjective treatment* could be more related to expectations rather than an actual reflection of the treatment benefits. This variation may explain the findings in study 2 (decrements in depression for subjective sham treatment) compared to study 1 (decrements in depression for subjective active treatment), where only in the former were subjects questioned during the procedure (week 1) and not at its end. This possibility is a post hoc explanation, and future experiments collecting data on participants' behavioural, cognitive and clinical outcomes should also record subjective expectations thoroughly (*Boot et al., 2013Braga et al., 2021*).

We want to highlight that, while we present *subjective treatment* as an important variable with explanatory power in addition to *objective treatment*, these results do not imply that participants' subjective beliefs can explain all of the variability in research outcomes (see also in Hochman et al., in preparation; commenting to *Gordon et al., 2022*). This is demonstrated in study 3, where the *objective treatment* significantly explained inattention symptoms even when *subjective treatment* was accounted for. Additionally, we present in Appendix 1 an example of a neuromodulation study in which *objective treatment* explained variability in treatment effects that could not be attributed to *subjective treatment* (*Murphy et al., 2020*). Based on this consideration, where researchers have data, examining variability in participants' *subjective treatment* may add further insight into prior results. However, unsurprisingly, when researchers were contacted about providing data on *subjective*

*treatment*, many reported that the assessment of subjective beliefs, aside from side effects, was not recorded. Indeed, even our group's procedure in the past lacked the recording of *subjective treatment* (e.g. *Looi and Cohen Kadosh, 2016*; *Cohen Kadosh et al., 2007*).

Overall, our findings hold twofold importance. Firstly, we introduce two new concepts in the academic literature: *subjective treatment* and *subjective dosage*. Secondly, we cast light on the role of participants' subjective experience in explaining the variability of results from RCTs and experiments that test the effectiveness of neuromodulation on mental health and behaviour. Altogether, we call for future studies to systematically collect data on participants' subjective beliefs and expectations. Studies that have collected data on subjective beliefs at the start and end of the intervention may consider examining the potential contribution of such beliefs to their results. Aside from estimating subjective beliefs about belonging to the active or control condition, we suggest that future research may consider collecting and analysing data on : (1) participants' beliefs before and at some midpoint during the experiment rather than only at the end, and (2) participants' expectations about the directionality and strength of the effect of *subjective treatment* on expected outcomes. This approach would be enhanced with designs that include deception, whereby all participants are told that they received the active treatment. However, such designs require careful ethical review, particularly in clinical populations. Overall, such data will allow a thorough examination of subjective beliefs, yielding more valid and replicable results to progress scientific and clinical studies to benefit human health and behaviour.

## Methods
### Participants and design
#### Study 1
One hundred and twenty-one patients (77 females, age range 18–85) with treatment-resistant depression took part in this study based on the data from *Blumberger et al., 2016*. Patients were randomised as part of a mixed design to receive sequential bilateral rTMS, unilateral high-frequency left (HFL)-rTMS, or sham rTMS for 3 or 6 weeks, depending on treatment response. Patients were included in the study if (1) the Structured Clinical Interview for DSM-IV provided a DSM-IV diagnosis of MDD; (2) they were experiencing a current major depressive episode with a score of 20 or higher on the 17-item HAMD-17; (3) they had failed to achieve a clinical response to or did not tolerate at least two different antidepressants from distinct classes at sufficient doses for at least 6 weeks; and (4) they had been receiving psychotropic medications for at least 4 weeks before randomisation took place. Patients were excluded if (1) a history of DSM-IV substance dependence was present in the 6 months before the study or a history of DSM-IV substance abuse was present in the month preceding the study; (2) the Structured Clinical Interview provided a DSM-IV provided a diagnosis of borderline personality disorder or antisocial personality disorder; (3) an unstable medical or neurological illness or a history of seizures was present; (4) they were suicidal; (5) they were pregnant; (6) had metal implants in the skull; (7) had a cardiac pacemaker; (8) had an implanted defibrillator or a medication pump; (9) presented a diagnosis of dementia or a current Mini-Mental State Examination (MMSE) score less than 24; and (10) they were taking lorazepam or an equivalent medication during the 4 weeks before the study.

#### Study 2
Fifty-two outpatients (20 females, age range 65–80) with late-life depression took part in this study, which was based on the data from *Kaster et al., 2018*. Patients were randomised as part of a mixed design to receive active deep rTMS or sham rTMS for 4 weeks. The same inclusion criteria applied as in experiment 1 aside from the age restriction and the depression diagnosis (defined based on a score of ≥22 on the HDRS-24). Similarly, in addition to the exclusion criteria outlined in experiment 1, patients were excluded if (1) any of the following diagnoses were present: bipolar I or II disorder, primary psychotic disorder, psychotic symptoms in the current episode, primary diagnosis of obsessive-compulsive, post-traumatic stress, anxiety, or personality disorder; (2) a dementia diagnosis was presented based on an MMSE with a score of <26; (3) rTMS contraindications (such as a history of seizures; intracranial implant); (4) a previously failed ECT trial during the current episode; (5) previous rTMS treatment; and (6) received bupropion >300 mg/day due to the dose-dependent increased risk of seizures.

### Study 3

Sixty-four patients (30 females, mean *M* = 38.6, *SD* = 9.6, age range 18-60) with ADHD (48% inattentive presentation and 52% combined presentation) took part in this study, which was based on the data from *Leffa et al., 2022*. Patients were randomised to receive active tDCS or sham tDCS for 4 weeks for a total of 28 sessions. Patients were included in the study if they (1) met DSM-5 criteria for ADHD based on a semistructured clinical interview conducted by trained psychiatrists; (2) were either not being treated with stimulants or agreed to perform a 30 day washout from stimulants before starting the tDCS; (3) estimated IQ score of 80 or above (based on Wechsler Adult Intelligence Scale, Third Edition); and (4) self-reported being of European descendant. Patients were excluded if they (1) showed moderate-to-severe symptoms of depression or depression based on BDI; (2) had a diagnosis of bipolar disorder with a manic or depressive episode or history of non-controlled epilepsy with seizures in the year prior to the study; (3) had a diagnosis of autism spectrum disorder or schizophrenia or psychotic disorder; (4) positive screened for substance use disorder, (5) showed unstable medical condition with reduction of functional capacity; (6) pregnancy or willingness to become pregnant in the 3 months subsequent to the beginning of the study; (7) inability to use the home-based tDCS device for any reason, and (8) previous history of neurosurgery or presence of any ferromagnetic metal in the head or implanted medical devices in the head or neck region. The outcome measure was based on the inattentive scores in the clinician-administered version of the Adult ADHD Self-report Scale version 1.1.

### Study 4

One hundred and fifty healthy participants (96 females, age *M* = 23, SD = 5) took part in this study, based on the data from *Filmer et al., 2019*. All subjects were right-handed, normal or corrected to normal vision, and passed a safety screening procedure. Participants were tested as part of a between-subject design. Subjects were randomly assigned to either one of the following five conditions: anodal 1 mA, cathodal 1 mA, 1.5 mA, 2 mA, or sham tDCS.

## Materials and procedure

### Study 1

All participants received treatment five times per week over 3 weeks for 15 treatments, only delivered on weekdays. After the first 3 weeks, participants were classified as either remitters (HAMD-17 score < 8) or non-remitters (HAMD-17 score ≥ 8) based on blinded clinical ratings. Those who achieved remission completed the study at week 3, while those classified as non-remitters entered a second phase, during which they received an additional 3 weeks of the same treatment under double-blind conditions. During the study, rTMS was administered using a Magventure RX-100 repetitive magnetic stimulator (Tonika/Magventure) and a cool B-65 figure-8 coil. To derive stimulation intensity, the motor threshold was obtained before treatment. In order to localise the stimulation site (left DLPFC), a structural MRI was coregistered to participants' heads using a magnetic tracking device (miniBIRD, Ascension Technology Group) for coil-to-cortex coregistration. Sham stimulation was administered in randomised fashion either as sham HFL-rTMS or sham bilateral rTMS with the coil angled 90° away from the skull in a single-wing tilt position, leading to some scalp sensations and sound intensity similar to that of active stimulation. Moreover, participants could not see the coil, reducing the likelihood of detecting the treatment allocation. Full details of the neuronavigation procedure and applied stimulation can be found in the supplementary material of *Blumberger et al., 2016*. After the final session, participants were asked whether they thought they received active or sham stimulation (presented as a binary choice).

### Study 2

Participants were randomised to active rTMS or sham rTMS, administered 5 d per week for a total of 20 treatments over 4 weeks, and only delivered during weekdays. Participants achieved remission by the end of week 4 (defined as both HDRS-24 ≤ 10 and ≥60% reduction from baseline on two consecutive weeks). Participants were withdrawn if HDRS-24 increased from baseline >25% on two consecutive assessments if they developed significant suicidal ideation or attempted suicide.

This study administered rTMS using a Brainsway deep rTMS system with the H1 coil device (Brainsway Ltd, Jerusalem, Israel). The intensity was derived using the resting motor threshold (RMT) obtained before treatment. All participants included in the analysis received rTMS with the H1 coil targeting the dorsolateral and ventrolateral prefrontal cortex bilaterally and performed at 120% of the RMT. The active rTMS group received the following standardised dose of rTMS: 18 Hz, at 120% RMT, 2 s pulse train, 20 s inter-train interval, 167 trains, for a total of 6012 pulses per session over 61 min. The sham group received treatment with the same parameters, device, and helmet. However, the active H1 coil was disabled when initiating the sham mode. A second coil (sham H1 coil) was located within the treatment helmet but activated far above the participant's scalp. This sham H1 coil delivered a tactile and auditory sensation similar to the active H1 coil, but the electric field was insufficient to induce neuronal activation. Full details regarding the applied stimulation can be found in *Kaster et al., 2018*. After the first session, participants were asked whether they thought they received active or sham stimulation (presented as a binary choice) via a short questionnaire.

## Study 3
The authors used a home-based tDCS device developed at Hospital de Clínicas de Porto Alegre for this study. The at-home tDCS device has been used in previous studies and included a user-friendly interface sensitive to impedance, such that sessions with too high impedance were automatically blocked. Furthermore, the number of the sessions, the dosage of the sessions and the stimulations were pre-programmed with a minimum interval between two consecutive sessions of 16 hr along with an option to abort a session (if necessary). Additionally, the capacity to save the number of sessions and time of stimulation performed by each participant was also controlled and pre-programmed. The current was delivered using 35 cm$^2$ electrodes (7 cm × 5 cm) coated with a vegetable sponge moistened with saline solution before the stimulation by two silicone cannulas coupled to the electrode. The electrodes were fixed on one of three sizes of neoprene caps that were given to each patient based on their head circumference.

Instructions on using the device were given at the baseline assessment when they received the first stimulation session, assisted by trained staff. Participants were instructed to remain seated during sessions, but no other behavioural restriction was imposed. Participants underwent 30 min daily sessions of tDCS, 2-mA direct constant current, for 4 weeks for a total of 28 sessions (including weekends). The anodal and cathodal electrodes were positioned over F4 and F3, corresponding to the right and left DLPFC according to the international 10–20 electroencephalography system. Devices programmed for sham treatment delivered a 30 s ramp-up (0–2 mA) stimulation followed by a 30 s ramp-down (2–0 mA) at the application's beginning, middle, and end. This procedure was performed to mimic the tactile sensations commonly reported with tDCS. Also, each participant received a daily reminder in the form of a text message on their cell phones to improve adherence. The participants were encouraged to perform the stimulation sessions at the same time of the day. At the end of the study, participants reported whether they thought they received active or sham stimulation (presented as a binary choice).

## Study 4
The experiment was conducted on a single day and consisted of three parts. Firstly, participants were familiarised with the experimental paradigm. Secondly, participants were instructed to sit quietly with their eyes open and stimulation was applied offline to the left prefrontal cortex for 20 min. Lastly, participants performed a sustained attention task for 40 min, during which mind wandering, the main outcome of this study, was measured. Overall, each participant completed a single session, lasting approximately 1.5 hr.

Stimulation was delivered with a NeuroConn stimulator (neuroConn GmbH, Ilmenau, Germany). The target was placed over F3 (EEG 10–20 system), and the reference was over the right orbitofrontal region. For the four groups who received active stimulation, tDCS lasted 20 min (including 30 s ramping up and down). During stimulation, participants were asked to sit quietly and keep their eyes open. The group that received sham stimulation had the same instructions but only received 15 s of constant current. The current was ramped up for 30 s up to 1.5 mA, then ramped down for 30 s. Stimulation was single-blinded, meaning that while the participants were blind to the stimulation they received, the experimenters were aware of the participant's stimulation group.

During the experiment, participants completed a sustained attention task (SART) in which they were asked to respond via a keypress (space bar) to non-target stimuli (single digits excluding the number 3) and withholding responses to target stimuli (the number 3) (see *Figure 3a*). Half of the trials ended in a target stimulus; the other half ended in a task-unrelated thought (TUT) probe. The TUT probe asked: "To what extent have you experienced task-unrelated thoughts prior to the thought probe? 1 (minimal) – 4 (maximal)". Participants' average response to the probe across trials was taken as a measure of mind wandering performance, with higher scores indicating higher mind wandering. At the end of the experiment, participants were asked whether they thought they received active or sham stimulation (presented as a binary choice) via a short questionnaire. Moreover, at the end of the study, participants were also asked to guess which stimulation dosage they received, choosing between the following options: none, weak, moderate, or strong.

## Statistical analysis

Statistical analysis was run using R (version 4.2.0. for Windows). When considering a dependent variable on a continuous scale (e.g. depression scores), the function *lme4* (*Bates et al., 2015*) was chosen to fit a linear mixed-effects model in the formulation described by *Laird and Ware, 1982*. This analytic framework has two advantages over non-mixed linear models: (1) it allows the pooling of the same grand mean for both sham and the active groups at the baseline, and (2) the within-group errors are allowed to be correlated and/or have unequal variances. Hence, the assumption of homoscedasticity can be violated. When the dependent variables were coded as binary (e.g. remission and response rates), the function *glm* was chosen to run general mixed-effects models.

We here refer to the subject's judgement of whether they received active or sham stimulation as *subjective treatment*, in opposition to *objective treatment*, which indicates the actual type of stimulation that each subject received during the experiment. Similarly, we refer to participants' judgement of stimulation dosage as *subjective dosage*. We performed a theoretically driven model comparison to address the following two questions: (1) Does the inclusion of *subjective treatment* lead to a model with a significantly better fit than the baseline model including *objective treatment* (and *time*, when applicable) and do they interact? (2) Does the inclusion of *objective treatment* lead to a model with a significantly better fit than the baseline model including *subjective treatment* (and *time*, when applicable)?

In order to address the first question, we defined a baseline model including *time*, and *time* by *objective treatment* interaction as fixed effects. Time was defined as a categorical variable, with each level reflecting the weekly assessments from baseline to the end of the study. Participants were entered into the model as random effects. Notably, the reference levels for all of the models (the intercepts) were the baseline; therefore, each effect was imposed as a difference compared to the baseline performance. Thus, the effect of time grasps the overall time difference compared to the baseline. In the same vein, the interaction terms of time and treatments (either subjective or objective) could be conceptualised as a covariate capturing the effect of the treatment over time when compared to the baseline performance. Given our interest in the contribution of *subjective treatment* over time, we compared the baseline model to an updated model that also included *subjective treatment* in a two-way interaction with *time*. Model comparison was run using the *anova* function in R (*R Development Core Team, 2022*). Our focus was on whether the comparison was significant at $\alpha < 0.05$, indicating that the inclusion of *subjective treatment* led to a considerably better model fit, explaining variability in the dependent variable in addition to the explanatory power of *objective treatment* over time.

Lastly, we compared the updated model to a more complex model, including the three-way interaction of time, *subjective treatment,* and *objective treatment*. In this case, our focus was whether the model comparison was significant, indicating that *subjective treatment* interacted with time and *objective treatment* to explain variability. As for the second question, we switched the order of the baseline models from the previous investigation. The baseline model included time and the interaction of *subjective treatment* with time, and then the compared model included *objective treatment*. Henceforth, the additional comparison of the three-way interaction was identical to the one in the first question. That allows for establishing if the *objective treatment* explained variability over *subjective treatment*.

## Additional information

### Funding

| Funder | Grant reference number | Author |
|---|---|---|
| Medical Research Council | RG86932 | Luisa Fassi |
| James S. McDonnell Foundation | | Roi Cohen Kadosh |
| National Institute of Mental Health | | Zafiris J Daskalakis |
| Canadian Institutes of Health Research | | Zafiris J Daskalakis Daniel M Blumberger |
| BrainsWay | | Zafiris J Daskalakis Daniel M Blumberger |
| Brain Canada | | Zafiris J Daskalakis |
| Temerty Family Foundation | | Zafiris J Daskalakis |
| Kreutzkamp Family Foundation | | Zafiris J Daskalakis |
| MagVenture Inc | | Zafiris J Daskalakis |

The funders had no role in study design, data collection and interpretation, or the decision to submit the work for publication.

### Author contributions

Luisa Fassi, Conceptualization, Data curation, Formal analysis, Investigation, Methodology, Writing – original draft, Writing – review and editing; Shachar Hochman, Formal analysis, Investigation, Methodology, Writing – original draft, Writing – review and editing; Zafiris J Daskalakis, Resources, Methodology, Writing – review and editing; Daniel M Blumberger, Conceptualization, Resources, Methodology, Writing – review and editing; Roi Cohen Kadosh, Conceptualization, Supervision, Investigation, Methodology, Project administration, Writing – review and editing

### Author ORCIDs

Luisa Fassi ⬤ https://orcid.org/0000-0002-0520-6425
Shachar Hochman ⬤ http://orcid.org/0000-0002-8322-3255
Daniel M Blumberger ⬤ https://orcid.org/0000-0002-8422-5818
Roi Cohen Kadosh ⬤ http://orcid.org/0000-0002-5564-5469

### Ethics

All four studies analysed complied with ethics standards and received approval from the ethics board of their respective institutions. The studies by Blumberger et al. (2016) and Kaster et al. (2018) were both approved by the CAMH Research Ethics Board. The study by Filmer et al. (2019) was approved by The University of Queensland Human Research Ethics Committee. The study by Leffa et al. (2022) was approved by The Research Ethics Committee of the Hospital de Clinicas de Porto Alegre.

Joint Public Review: https://doi.org/10.7554/eLife.88889.3.sa1
Author Response https://doi.org/10.7554/eLife.88889.3.sa2

## Additional files

### Supplementary files
• MDAR checklist

### Data availability

This study includes analyses on four datasets (*Blumberger et al., 2016*; *Kaster et al., 2018*; *Leffa et al., 2022*; *Filmer et al., 2019*). The dataset by *Filmer et al., 2019* was originally open at: https://

espace.library.uq.edu.au/view/UQ:3b30c6d. We made the de-identified datasets from *Blumberger et al., 2016*, *Kaster et al., 2018* and *Leffa et al., 2022* available on the Open Science Framework (OSF) upon approval from the study authors. The data and the R code can be found on the OSF at: https://osf.io/rztxu/.

The following dataset was generated:

| Author(s) | Year | Dataset title | Dataset URL | Database and Identifier |
|---|---|---|---|---|
| Hochman S, Fassi L, Cohen Kadosh R | 2024 | The Importance of Accounting for Participants' Subjective Beliefs When Assessing Outcomes in Neurostimulation Treatments | https://osf.io/rztxu/ | Open Science Framework, 10.17605/OSF.IO/RZTXU |

The following previously published dataset was used:

| Author(s) | Year | Dataset title | Dataset URL | Database and Identifier |
|---|---|---|---|---|
| Filmer H, Griffin A, Dux PE | 2019 | Dosage dependent increases in mind wandering via prefrontal tDCS | https://doi.org/10.14264/uql.2019.295 | UQ eSpace, 10.14264/uql.2019.295 |

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

## Appendix 1

We here apply our approach to experimental study that examined the effect of non-invasive brain stimulation (NIBS) on participants' working memory. In this case, we show that the inclusion of *subjective treatment* did not explained variability in experimental outcomes.

This study (*Murphy et al., 2020*) compared the effects of different NIBS techniques, namely anodal tDCS, tRNS + DC-offset, or sham stimulation over the left DLPFC on working memory (WM) performance and task-related EEG oscillatory activity in 49 healthy adults. Participants were allocated to receive either anodal tDCS (N = 16), high-frequency tRNS + DC-offset (N = 16), or sham stimulation (N = 17) to the left DLPFC using a between-subjects design. The Sternberg WM task was used to measure changes in WM performance before, 5 min and 25 min post-stimulation. Moreover, event-related synchronisation/desynchronisation (ERS/ERD) of oscillatory activity was analysed from EEG recorded during WM encoding and maintenance. At the end of the experiment, participants were asked whether they thought they received active or sham stimulation (presented as a binary choice) via a short questionnaire.

We employed the same analytical approach as for experiments 1–4, looking at the contribution of *subjective treatment* to model fit over and beyond *objective treatment* and examining whether *objective treatment* explained variability in experimental outcomes (WM accuracy and reaction times) over and beyond *subjective treatment*. A linear mixed model with WM reaction time was fitted to the data for time 0–2 (before NIBS, 5 min after, and 25 min after). The baseline model included *objective treatment* (sham/tRNS/tDCS) as the main effect, as well as the interaction of *time* and *objective treatment*. We first added to this model *subjective treatment* (active or placebo) as a main effect. Next, we extended the model to include the two-way interaction of *time* and *subjective treatment*. Lastly, we considered a model with the three-way interaction of *time*, *subjective treatment,* and *objective treatment*. Our results showed that the inclusion of *subjective treatment* did not lead to a better model fit neither as a main effect (*df* = 13, *BIC* = 1824.16, *AIC* = 1785.28, *p* = 0.670) nor as a two-way interaction with *time* (*df* = 15, *BIC* = 1833.25, *AIC* = 1788.40, *p* = 0.642) and a three-way interaction with *time* and *objective treatment* (*df* = 21, *BIC* = 1856.48, *AIC* = 1793.70, *p* = 0.348). Hence, participants' subjective experience about the treatment did not explain variability in reaction times beyond the actual treatment condition to which participants were assigned.

We next examined whether the changes in reaction time were explained by *objective treatment*. We, therefore, added *objective treatment* first and, following, *objective treatment * time* after *subjective treatment * time* was already included in the baseline model. Our results showed that the inclusion of neither *objective treatment* (*df* = 11, *BIC* = 1820.58, *AIC* = 1787.69, *p* = 0.139) nor *objective treatment * time* (*df* = 15, *BIC* = 1833.25, *AIC* = 1788.40, *p* = 0.121) led to a better model fit. Therefore, when accounting for participants' subjective experience of receiving the real or placebo treatment, the actual treatment to which participants were assigned during the experiment did not contribute to explaining changes in WM reaction time.

Following, we run the same analysis with WM accuracy as the outcome of interest. Model comparison showed that *subjective treatment* did not lead to a better model fit neither as a main effect (*df* = 13, *BIC* = 1084.64, *AIC* = 1045.77, *p* = 0.069) nor as a two-way interaction with *time* (*d f* = 15, *BIC* = 1094.60, *AIC* = 1049.74, *p* = 0.990) and a three-way interaction with *time* and *objective treatment* (*df* = 21, *BIC* = 1122.48, *AIC* = 1059.68, *p* = 0.914). On the contrary, we next examined whether the changes in WM accuracy were explained by *objective treatment*. Our results showed that the inclusion of *objective treatment* (*df* = 11, *BIC* = 1089.73, *AIC* = 1056.84, *p* = 0.183) did not lead to a better model fit, but the interaction of *objective treatment * time* did (*df* = 15, *BIC* = 1094.60, *AIC* = 1049.74, *p* = 0.005). Therefore, when accounting for participants' subjective experience of receiving the real or placebo treatment, the actual treatment to which participants were assigned during the experiment contributed to explaining changes in WM accuracy.

