## [Editor Report · eLife assessment]

This is an **important** report that has implications for both the brain stimulation field and beyond. The strength of evidence provided is quite **convincing**. The major strength of this work is the recognise the importance of participant expectation in brain stimulation studies.

---

## [Referee Report · Joint Public Review]

Randomized clinical trials use experimental blinding and compare active and placebo conditions in their analyses. In this study, Fassi and colleagues explore how individual differences in subjective treatment (i.e., did the participant think they received the active or placebo treatment) influence symptoms and how this is related to objective treatment. Authors address this highly relevant and interesting question using a powerful method by (re-)analyzing data from four published neurostimulation studies and including subjective treatment in statistical models explaining treatment response. The major strengths include the innovative and important research question, the inclusion of four different studies with different techniques and populations to address this question, sound statistical analyses, and findings that are of high interest and relevance to the field.

The paper will have significant impact on the field. It will promote further investigation of the effects of sham vs active treatment by the introduction of the terms subjective treatment vs objective treatment and subjective dosage that can be used consistently in the future. The suggestions to assess the expectation of sham vs active earlier on in clinical trials will advance the understanding of subjective treatment in future studies. Overall, I believe the data will substantially contribute to the design and interpretation of future clinical trials by underscoring the importance of subjective treatment.

---

## [Author Response]

The following is the authors’ response to the original reviews.

**Reviewer 1**

**Public Review**
(R1.1) Randomized clinical trials use experimental blinding and compare active and placebo conditions in their analyses. In this study, Fassi and colleagues explore how individual differences in subjective treatment (i.e., did the participant think they received the active or placebo treatment) influence symptoms and how this is related to objective treatment. The authors address this highly relevant and interesting question using a powerful method by (re-)analyzing data from four published neurostimulation studies and including subjective treatment in statistical models explaining treatment response. The major strengths include the innovative and important research question, the inclusion of four different studies with different techniques and populations to address this question, sound statistical analyses, and findings that are of high interest and relevance to the field.

We thank the reviewer for this summary and the overall appreciation for our work.

(R1.2) My main suggestion is that authors reconsider the description of the main conclusion to better integrate and balance all findings. Specifically, the authors conclude that (e.g., in the abstract) "individual differences in subjective treatment can explain variability in outcomes better than the actual treatment", which I believe is not a consistent conclusion across all four studies as it does not appropriately consider important interactions with objective treatment observed in study 2 and 3. In study 2, the greatest improvement was observed in the group that received TMS but believed they received sham. While subjective treatment was associated with improvement regardless of objective active or sham treatment, improvement in the objective active TMS group who believed they received sham suggests the importance of objective treatment regardless of subjective treatment. In Study 3, including objective treatment in the model predicted more treatment variance, further suggesting the predictive value of objective treatment.

We thank the reviewer for this comment and agree that the interpretation of findings requires a more nuanced and balanced description. We, therefore, implemented changes in both the abstract and discussion of the manuscript, as reported below (additions are highlighted in grey and deletions are shown in strikethrough):

Abstract

“Our findings consistently show that the inclusion of subjective treatment can provide a better model fit when accounted for alone or in an interaction term with objective treatment (defined as the condition to which participants are assigned in the experiment). These results demonstrate the significant contribution of subjective experience in explaining the variability of clinical, cognitive and behavioural outcomes. Based on these findings, We advocate for existing and future studies in clinical and non-clinical research to start accounting for participants’ subjective beliefs and their interplay with objective treatment when assessing the efficacy of treatments. This approach will be crucial in providing a more accurate estimation of the treatment effect and its source, allowing the development of effective and reproducible interventions.” (p. 3)

Discussion

“We demonstrate that participants’ subjective beliefs about receiving the active vs control (sham) treatment are an important factor that can explain variability in the primary outcome and, in some cases, fits the observed data better than the actual treatment participants received during the experiment.” (p. 21)

“We demonstrate that participants’ subjective beliefs about receiving the active vs control (sham) treatment are an important factor that can explain variability in the primary outcome and, in some cases, fits the observed data better than the actual treatment participants received during the experiment. Specifically, in Studies 1, 2 and 4, the fact that participants thought to be in the active or control condition explained variability in clinical and cognitive scores to a more considerable extent than the objective treatment alone. Notably, the same pattern of results emerged when we replaced subjective treatment with subjective dosage in the fourth experiment, showing that subjective beliefs about treatment intensity also explained variability in research results better than objective treatment. In contrast to Studies 1 and 4, Studies 2 and 3 showed a more complex pattern of results. Specifically, in Study 2 we observed an interaction effect, whereby the greatest improvement in depressive symptoms was observed in the group that received the active objective treatment but believed they received sham. Differently, in Study 3, the inclusion of both subjective and objective treatment as main effects explained variability in symptoms of inattention. Overall, these findings suggest the complex interplay of objective and subjective treatment. The variability in the observed results could be explained by factors such as participants’ personality, type and severity of the disorder, prior treatments, knowledge base, experimental procedures, and views of the research team, all of which could be interesting avenues for future studies to explore.” (p. 22)

(R1.3) In addition to updating the conclusions to better reflect this interaction, I suggest authors include the proportion of participants in each subjective treatment group that actually received active or sham treatment to better understand how much of the subjective treatment is explained by objective treatment. I think it is particularly important to better integrate and more precisely communicate this finding, because the conclusions may otherwise be erroneously interpreted as improvements after treatment only being an effect of subjective treatment or sham.

We thank the reviewer for this comment. The information about how many participants are included in each group is provided in the every each codebooks under the section “Count of Participants by Treatment Condition and Their Subjective Guess” which is in the project’s OSF link (https://osf.io/rztxu/). Additionally, we added these tables to the supplementary material in tables S1, S8, S15, and S18, and we referred to these tables throughout the Methods section. Further, we added this information to the manuscript results, as follows:

“Further details on participant groupings based on objective treatment and their subjective treatment can be found in the codebook corresponding to each of the four studies as well as S1.” (p. 8).“The breakdown of participants to objective treatment and subjective treatment in the sample can be found in S8.” (p. 13).“The breakdown of participants to objective treatment and subjective treatment in the sample can be found in S15.” (p. 17).“The breakdown of participants to objective treatment and subjective treatment in the sample can be found in S18.” (p. 19).

(R1.4) The paper will have significant impact on the field. It will promote further investigation of the effects of sham vs active treatment by the introduction of the terms subjective treatment vs objective treatment and subjective dosage that can be used consistently in the future. The suggestions to assess the expectation of sham vs active earlier on in clinical trials will advance the understanding of subjective treatment in future studies. Overall, I believe the data will substantially contribute to the design and interpretation of future clinical trials by underscoring the importance of subjective treatment.

We thank the reviewer for this positive comment.

**Review for authors**
(R1.4) Abstract"Here we show that individual differences in subjective treatment.. can explain variability in outcomes better than the actual treatment". "Our findings consistently show that the inclusion of subjective treatment provides a better model fit than objective treatment alone" - these two statements could be interpreted as two different conclusions, authors should be more consistent.

We thank the reviewer for this comment and have now changed the abstract to be consistent, as also highlighted in R1.1:

Abstract

“Our findings consistently show that the inclusion of subjective treatment can provides a better model fit when accounted for alone or in an interaction term with objective treatment (defined as the condition to which participants are assigned in the experiment). These results demonstrate the significant contribution of subjective experience in explaining the variability of clinical, cognitive and behavioural outcomes. Based on these findings, We advocate for existing and future studies in clinical and non-clinical research to start accounting for participants’ subjective beliefs and their interplay with objective treatment when assessing the efficacy of treatments. This approach will be crucial in providing a more accurate estimation of the treatment effect and its source, allowing the development of effective and reproducible interventions.” (p. 3)

(R1.5) IntroductionThis is an odd sentence given it is 2023: "As a result, the global neuromodulation device industry is expected to grow to $13.3 billion in 2022 (Colangelo, 2020)."

We have now removed this sentence as indeed not applicable and instead added a reference for the previous sentence:

“In recent years, neuromodulation has been studied as one of the most promising treatment methods (De Ridder et al., 2021).”

Reference

De Ridder, D., Maciaczyk, J., & Vanneste, S. (2021). The future of neuromodulation: Smart neuromodulation. Expert Review of Medical Devices, 18(4), 307–317.https://doi.org/10.1080/17434440.2021.1909470

(R1.6) FiguresLines of Figure 1 are vague.Figure 5 color scheme is confusing. It would be better to use green/blue colors for one, (e.g.) sham in both subjective and objective treatment and orange/red colors for active treatment.For Figure 6 it would be better to use the same color for sham as subjective dosage none.Relatedly, it would be easier to keep color scheme consistent across the paper and for example use green/blue colors for sham throughout.

We thank the reviewer for this comment. Following these comments, all the figures of the paper has remade for better clarity.

Figure 1, the individual lines are now shown stronger, there is also a connecting line between the averages.Figure 5, sham is now on cold colours (blue and green), and active treatment on warm colours (red and orange)Figure 6, the same colour for sham as subjective dosage none is now applied.

Further, we also edited Figures 2 and 4 by removing the percentages between 0% and 100% on the y-axis. Given that the outcome variable was binary coded, we implemented this change to avoid confusion.

**Reviewer 2**

**Public Review**
(R2.1) This manuscript focuses on the clinical impact of subjective experience or treatment with transcranial magnetic stimulation and transcranial direct current stimulation studies with retrospective analyses of 4 datasets. Subjective experience or treatment refers to the patient level thought of receiving active or sham treatments. The analyses suggest that subjective treatment effects are an important and under appreciated factor in randomized controlled trials. The authors present compelling evidence that has significance in the context of other modalities of treatment, treatment for other diseases, and plans for future randomized controlled trials. Other strengths included a rigorous approach and analyses. Some aspects of the manuscript are underdeveloped and the findings are over interpreted. Thank you for your efforts and the opportunity to review your work.

We thank the reviewer for their overall appreciation of this work. We address the comment on the overinterpretation of findings in response to reviewer 1 (see R1.2) above, and we expand on the underdeveloped explanation of sham procedures (see R2.2) below.

**Review for authors**
(R2.2) One concern is that the findings are consistently over interpreted and presented with a polarizing framework. This is a complicated area of study with many variables that are not understood or captured. For example, subjective experience effects likely varies with personality dimensions, disease, prior treatments, knowledge base, view of the research team, and disease severity. Framing subjective experience with a more balanced tone, as an important consideration for future trial design and study execution would enhance the impact of the paper.

We thank the reviewer for this comment. We reframed our interpretation of results in both the manuscript abstract and discussion, as highlighted in response to reviewer 1 (see R1.2) above.

(R2.3) The discussion of sham approaches for transcranial magnetic stimulation and transcranial direct current stimulation is underdeveloped. There are approaches that are not discussed. The tilt method is seldom used for modern studies for example.

We thank the reviewer for this comment, and we now rewrote a paragraph elaborating more on different practices to apply sham procedures in the introduction section:

“Participants that take part in TMS and tES studies consistently report various perceptual sensations, such as audible clicks, visual disturbances, and cutaneous sensations (Davis et al., 2013) Consequently, they can discern when they have received the active treatment, making subjective beliefs and demand characteristics potentially influencing performance (Polanía et al., 2018). To account for such non-specific effects, sham (placebo) protocols have been employed. For transcranial direct current stimulation (tDCS), the most common form of tES, various sham protocols exist. A review by Fonteneau et al., 2019 shows 84% of 173 studies used similar sham approaches to an early method by Gandiga et al., 2005. This initial protocol had a 10s ramp-up followed by 30s of active stimulation at 1mA before cessation, differently from active stimulation that typically lasts up to 20 minutes.. However, this has been adapted in terms of intensity and duration of current, ramp-in/out phases, and the number of ramps during stimulation. Similarly, in sham TMS, the TMS coil may be tilted or replaced with purpose-built sham coils equipped with magnetic shields, which produce auditory effects but ensure no brain stimulation (Duecker & Sack, 2015). By using surface electrodes, the somatosensory effects of actual TMS are also mimicked. Overall, these types of sham stimulation aim to mimic the perceptual sensations associated with active stimulation without substantially affecting cortical excitability (Fritsch et al., 2010; Nitsche & Paulus, 2000). As a result, sham treatments should allow controlling for participants’ specificbeliefs about the type of stimulation received.” (p.6)

References

Fonteneau, C., Mondino, M., Arns, M., Baeken, C., Bikson, M., Brunoni, A. R., Burke, M. J., Neuvonen, T., Padberg, F., Pascual-Leone, A., Poulet, E., Ruffini, G., Santarnecchi, E., Sauvaget, A., Schellhorn, K., Suaud-Chagny, M.-F., Palm, U., & Brunelin, J. (2019). Sham tDCS: A hidden source of variability? Reflections for further blinded, controlled trials. Brain Stimulation, 12(3), 668–673. https://doi.org/10.1016/j.brs.2018.12.977

Gandiga, P. C., Hummel, F. C., & Cohen, L. G. (2006). Transcranial DC stimulation (tDCS): A tool for double-blind sham-controlled clinical studies in brain stimulation. Clinical Neurophysiology, 117(4), 845–850. https://doi.org/10.1016/j.clinph.2005.12.003